# Effects of Ovine Monocyte-Derived Macrophage Infection by Recently Isolated *Toxoplasma gondii* Strains Showing Different Phenotypic Traits

**DOI:** 10.3390/ani12243453

**Published:** 2022-12-07

**Authors:** Raquel Vallejo, Julio Benavides, Noive Arteche-Villasol, Mercedes Fernández-Escobar, María Del Carmen Ferreras, Valentín Pérez, Daniel Gutiérrez-Expósito

**Affiliations:** 1SANPATRUM, Animal Health Department, University of León, Campus de Vegazana s/n, 24071 León, Spain; 2Instituto de Ganadería de Montaña (CSIC-ULE), Grulleros, 24346 León, Spain; 3SALUVET, Animal Health Department, Faculty of Veterinary Sciences, Complutense University of Madrid, Ciudad Universitaria s/n, 28040 Madrid, Spain

**Keywords:** *Toxoplasma gondii*, sheep, genotype, macrophages, internalization, cytokines

## Abstract

**Simple Summary:**

Toxoplasmosis is a zoonotic disease caused by a protozoan parasite called *Toxoplasma gondii*. This parasite affects all warm-blooded animals. It is one of the more common causes of abortions in sheep, which threatens both the welfare of the animals and the economic sustainability of the farms. The virulence of different *T. gondii* isolates have been studied in mice and humans, but little information is available regarding sheep. The aim of this work was to study how the genetic variability of six *T. gondii* strains, that were recently isolated from Spanish sheep, affected different phenotypic traits in an in vitro model using ovine monocyte-derived macrophages. Our results showed that the type III isolates had a higher internalization/infection rate than type II isolates. Moreover, these two isolates also were shown to have higher increments in cytokines that favored inflammation and a Th1 immune response. The results of this study differ with earlier findings in mouse models and in vitro investigations, as well as from results between the same genotypes and different genotypes; proving that more phenotypic traits are needed for the study of the virulence of *T. gondii* isolates.

**Abstract:**

Ovine toxoplasmosis is one the most relevant reproductive diseases in sheep. The genetic variability among different *Toxoplasma gondii* isolates is known to be related to different degrees of virulence in mice and humans, but little is known regarding its potential effects in sheep. The aim of this study was to investigate the effect of genetic variability (types II (ToxoDB #1 and #3) and III (#2)) of six recently isolated strains that showed different phenotypic traits both in a normalized mouse model and in ovine trophoblasts, in ovine monocyte-derived macrophages and the subsequent transcript expression of cytokines and iNOS (inducible nitric oxide synthase). The type III isolate (TgShSp24) showed the highest rate of internalization, followed by the type II clonal isolate (TgShSp2), while the type II PRU isolates (TgShSp1, TgShSp3, TgShSp11 and TgShSp16) showed the lowest rates. The type II PRU strains, isolated from abortions, exhibited higher levels of anti-inflammatory cytokines and iNOS than those obtained from the myocardium of chronically infected sheep (type II PRU strains and type III), which had higher levels of pro-inflammatory cytokines. The present results show the existence of significant intra- and inter-genotypic differences in the parasite-macrophage relationship that need to be confirmed in in vivo experiments.

## 1. Introduction

*Toxoplasma gondii* (Apicomplexa) is a zoonotic protist that can infects almost all warm-blooded animals. It is one of the main causes of reproductive failure and is responsible for important economic losses in the ovine industry [1,2]. As an obligate intracellular parasite, *T. gondii* can infect any kind of nucleated cell, including muscle, nervous, epithelial or immune cells, where it multiplies via endodyogeny [3,4]. Macrophages (MØs) are the primary target cells of *T. gondii* and play a key role in the host’s immune response against it. Their involvement in the pathogenesis of toxoplasmosis has been investigated [5,6,7]. The interaction of *T. gondii* with the different cells of the innate and adaptive immune response can lead to the production of Th1 cytokines (e.g., IL-12 and IFN-γ) that induce the type M1 phenotype of macrophages or classical activation, in which they show enhanced antimicrobial effects against *T. gondii* but, collaterally, may also cause tissue damage due to the pro-inflammatory immune response [7,8,9]. On the other hand, macrophages can be alternatively activated (M2 phenotype) by Th2 type cytokines (e.g., IL-4, IL-10) which leads to an anti-inflammatory response, reduces their ability to kill intracellular parasites and activates their role in tissue repair [10,11].

The balance between each type of immune response determines the outcome of the infection and it can be influenced, by the genetics of the isolate/strain, among other variables [12]. In fact, isolate differences in pathogenicity have been shown in different hosts [13]. In this sense, most *T. gondii* isolates belong to three main clonal genotypes: the highly virulent type I, moderately virulent type II, and non-virulent type III [14,15], each characterized according to their virulence (mortality rate) shown in murine models. However, this simplistic classification deserves further investigation because the current global perspective seems to be insufficiently clear to draw robust conclusions [16]. In addition, *T. gondii* virulence is also determined by several in vitro phenotypic traits such as invasiveness, replication capability, and the interaction with immune cells [16,17]. Regarding the genetic variability of *T. gondii* that infect sheep, the type II isolates are predominant among European flocks, followed in frequency by type III and recombinant isolates [18]. 

Multiple in vitro studies carried out with primary macrophages or with cell lines based on immortalized MØs from mice and humans have shown that the immune response of these cells after infection is dependent on the *T. gondii* isolate’s genotype [17,19,20,21]. Macrophages infected with type I and III isolates are expected to be alternatively (M2) activated, while macrophages infected with type II are classically (M1) activated [12]. Despite extensive research on the interaction between *T. gondii* and macrophages, little is known about ovine MØs (OvMØs), and scant information exists on how the genetic background or even the origin of the different isolates impacts innate immune parameters in an isolate-specific manner in these ovine target cells. Furthermore, recent studies on ovine trophoblast cells have shown that there is significant variation in virulence traits that cannot be inferred by genetic characterization using the currently described molecular markers [22]. In addition, the role of other unexplored characteristics of each isolate such as the origin of the clinical sample (e.g., aborted fetal brain tissues, adult sheep myocardium, and oocysts from cats) also needs to be considered.

Based on previously published results of in vivo and in vitro assays for the phenotypic characterization of several *T. gondii* isolates [22], a total of six strains belonging to different genotypes, geographical locations and origins of tissue sample and recently isolated from sheep, were selected with the intention of evaluating, through in vitro assays with OvMØs, the effect of such features by studying the cell infection rate (cIR) using immunostaining and the immune response through transcript expression of cytokines and iNOS after infection.

## 2. Materials and Methods

### 2.1. Ethics Statement

Sheep handling and blood-sample collection were carried out in accordance with the European Union legislation (Law 6/2013) concerning animals and their exploitation, transportation, experimentation and sacrifice; R.D. 118/2021 for the protection of animals employed in research and teaching; and Directive 2010/63/UE related to the protection of animals used for scientific goals. All procedures were approved by the local government (Junta de Castilla y León) after a positive report from the ethics committee of the Spanish Research Council (Ref. 100102/2018-6). All animals used in this study were handled in strict accordance with good clinical practices, and all efforts were made to minimize suffering.

### 2.2. In Vitro Generation of Ovine Monocyte-Derived Macrophages

The generation of the OvMØs was carried out as previously described [23]. Briefly, 500 mL of whole blood was collected in a blood-bag system with citrate phosphate dextrose adenine-1 (CPDA-1) (Teruflex^®^; Terumo, Tokio, Japan) from a healthy 2-year-old sheep housed at the Instituto de Ganadería de Montaña (CSIC-Universidad de León), León, Spain. The sheep was seronegative for *T. gondii* (ID Screen^®^ Toxoplasmosis indirect multi-species ELISA test; IDVet, Grabels, France) and *Neospora caninum* (ID Screen^®^
*Neospora caninum* indirect multi-species ELISA test; IDVet, Grabels, France). The same sheep was used for the entire study, and the blood samples were taken every two months. Peripheral blood mononuclear cells (PBMCs) were separated via gradient density centrifugation on Lymphoprep™ (STEMCELL Technologies^®^; Cologne, Germany) and seeded in cell-culture flasks (GRYNIA^®^; Labbox, Barcelona, Spain) at a density of 10^7^/mL and cultured in supplemented RPMI1640 medium (Gibco™; Thermo Fisher Cientifict, Paisley, UK) [23]. After a 3-h incubation at 37 °C and 5% CO_2_, non-adherent cells (lymphocytes) were removed, and adherent cells (monocytes) were incubated in RPMI medium supplemented with 80 ng/mL of ovine GM-CSF (RP1190V-100, KingFisher Biotech^®^; Saint Paul, MN, USA). After 3 days, the flasks were washed twice with PBS and the medium was replaced with fresh supplemented medium with GM-CSF. At 7 days of culture, the OvMØs were harvested and counted in a Neubauer chamber (BRAND^®^; Wertheim, Germany), and the viability was checked using trypan blue stain, which was usually above 95%. The identity and purity of the OvMØs was previously confirmed using a cytometry analysis [23]. Prior to parasite infection, a total of 2 × 10^5^ OvMØs in 1 mL of supplemented medium without GM-CSF were seeded in each well of a 24-well plate for 24 h. All assays were carried out in triplicate with the same conditions in three different experiments and with two duplicates for each experiment.

### 2.3. Toxoplasma Gondii Cultures and OvMØs Infection

For the in vitro assays, a panel of six Spanish *T. gondii* isolates that were recently obtained from ovine fetal abortions (*n* = 3) (TgShSp1, TgShSp2, and TgShSp3) or the myocardium of chronically infected sheep (*n* = 3) (TgShSp11, TgShSp16, and TgShSp24) were selected based on the previous results of genetic and phenotypic characterizations (Table 1) [22,24].

All isolates used in this study had a restricted number of passages completed in a MARC–145 cell line (from 8 to 15) to avoid an adaptation to the cell culture [25,26]. The tachyzoites were routinely maintained in a MARC–145 cell line as previously described [27]. 

The *Toxoplasma gondii* tachyzoites used for the OvMØs infection assays were harvested 4 days after infecting the MARC–145 cell cultures when most of the parasites were forming intracellular parasitophorous vacuoles inside the cells and scant tachyzoites were extracellular. They were passed through a 25 G needle and purified throughout PD–10 desalting columns (Cytiva^®^, Amersham Biosciences, Uppsala, Sweden) as previously described [28]. The viability was confirmed via trypan blue stain and the number of viable tachyzoites was determined by counting in a Neubauer chamber. Afterward, the *T. gondii* tachyzoites were inoculated at a multiplicity of infection (MOI) of 3 into a monolayer of 2 × 10^5^ OvMØs within the first hour after parasite purification and then incubated at 37 °C and 5% CO_2_ in a humidified chamber for 6 h. After incubation, the OvMØs were washed twice with warmed PBS to carry out different assays. Non-infected OvMØs were used as negative controls.

### 2.4. Cell Infection Rate and Multi-Infected Cell Rate

The OvMØs were previously seeded onto a sterile glass slide of 13 mm ø (VWR^®^ microscope cover glasses) to carry out the cell infection rate (cIR) assays, which were defined as the percentage of cells infected with one or multiple tachyzoites. The multi-infected cell rate (McR) was obtained as the percentage of multi-infected cells within the total of infected cells. Specifically, in order to investigate early interactions in the OvMØs after challenging with *T. gondii*, the time after infection for collecting samples was chosen based on previous studies of *Mycobacterium tuberculosis* and *Eimeria* spp. [29,30] and the authors’ previous in vitro observations of *T. gondii* at 1, 2, 4, 6, and 24 h after infection (hpi), in which at 6 hpi the OvMØs showed an elevated cIR (50%) (Appendix A). Additionally, regarding cytokine expression, we were interested in studying early immune response for which 6 hpi were more accurate than 24 hpi. Once established, the MOI was also chosen following previous studies on *T. gondii* involving macrophages from mice and rats [31,32] on *Neospora caninum* in bovine MØs where 3:1 was used [33], and authors previous assays with *T. gondii* and different MOIs (1, 3 and 5), which resulted in an infection rate higher than 50% using a MOI of 3 (Appendix A). 

### 2.5. Immunofluorescence Staining and Image Analysis

The OvMØs attached in glass slides were fixed and permeabilized with 500 µL of methanol for 20 min at −20 °C, washed twice with PBS, and stored with 2 mL of PBS per well at 4 °C until immunostaining as previously described [34]. Briefly, the tachyzoites data not shown were labelled using a 1:50 dilution of primary polyclonal antibody against *T. gondii* (220A-15-RUO, Cellmarque^®^, Rocklin, CA, USA) in animal-free blocker and diluent (Vector laboratories) and incubated overnight at 4 °C in a humidified chamber. Afterward, a goat anti-rabbit antibody conjugated to Alexa Fluor^®^ 488 (ab150077, Abcam; Cambridge, United Kingdom) was added as a secondary antibody (1:2000 in blocking buffer) for 45 min at room temperature. Then, the cover glasses were washed with PBS and incubated at 1:500 with Cellmask^TM^ (Thermo Fisher Scientific) labelled with tetramethylrhodamine (TRITC) for 30 min in complete darkness for the identification of the cytoplasm. Finally, coverslips were mounted on glass slides (HDMED^®^ cover glasses; Dossenheim, Germany) with 4′,6-diamidino-2-phenylindole (DAPI) mounting solution (Abcam, Cambridge, UK). OvMØs that had not undergone infection were included as negative controls to eliminate potential fluorescent artifacts.

Cell counting was carried out with digital photographs from 10 randomly selected fields per cover glass using an inverted fluorescence microscope (Eclipse Ni-E, Nikon) and a CMOS scientific camera (Photometrics^®^ Prime BSI™, Tucson, AZ, USA) with three different filters to visualize the tachyzoites, cytoplasm, and nuclei stained with Alexa Fluor^®^ 488, Cellmask^TM^ (ThermoFisher Scientific, Waltham, MA, USA), and DAPI, respectively, at a magnification of 200×. The images were merged using NIS-Elements (Nikon; Melville, NY, USA) software and analyzed using FIJI (ImageJ) [35]. The total number of OvMØs in each image was counted, resulting in a mean of 97 cells/field, and then they were classified according to the number of tachyzoites inside the cells. Then, the cIR and McR were calculated. Tachyzoites that presented altered morphology were not considered; only intact and whole cells were counted. The count was carried out independently by two operators (R.V. and J.B.) to avoid subjectivity.

The precision and accuracy of the analytical method used were evaluated by performing six replicates of each isolate (two for each experiment) with the same conditions. The final values of the cIR and McR were obtained by means of the average value of the six replicates performed in the three experiments. Bearing in mind that the presence of intracytoplasmic tachyzoites could be based either on actin-filament-dependent phagocytosis or active invasion, the term internalization was used when referring to both situations. The intra-experiment and inter-experiment precision was expressed as the average standard deviation (SD) within each isolate.

### 2.6. Analysis of Cytokine and iNOS Expression

The mRNA expression levels were determined via quantitative real-time PCR (qPCR) as previously described [23]. Primers for IFN-γ, IL-12, IL-10, IL-4, IL-6, IL-17, TNF-α, and TGF-β cytokines, as well as iNOS and the housekeeping gene β-actin, were used as published [23,36]. Additionally, the IL-1α primers were designed and checked by using Primer3Plus and Oligoanalyzer Tool software [37]. All primer sequences are listed in Appendix A.

After incubation, the supernatants were removed and the OvMØs were washed with PBS and collected for RNA extraction to evaluate the mRNA expression by adding 200 µL of RNeasy Mini Kit lysis buffer (RNeasy^®^ Mini Kit, Quiagen; Hilden, Germany) per well [38] and stored at −80 °C for subsequent RNA isolation. The total RNA isolation from the OvMØs was carried out using a commercial kit (RNeasy^®^ Mini Kit, Quiagen, Hilden, Germany) following the manufacturer´s recommendations. The RNA concentrations were determined using a QuantiFluor™ RNA System kit (Promega^®^; Madison, WI, USA) and a Quantus™ fluorimeter (Promega^®^, Madison, WI, USA). The integrity of the RNA samples was checked via the 260/280 absorbance ratio (samples for reverse transcription were chosen only when the values were close to 2.0) and after electrophoresis on a 1% agarose gel to confirm the adequate integrity of the 18S and 28S ribosomal subunits. The cDNA was obtained via reverse transcription using SuperScript™ VILO cDNA Master Mix (Invitrogen™, Thermo Fisher, Paisley, UK) in a 20 µL reaction up to 2.5 µg of total RNA and run in a SimpliAmp™ thermal cycler (Applied Biosystems™, Warrington, UK). The obtained cDNA was diluted up to 10 ng/µL with nuclease-free water, stored at −80 °C, and analyzed in qPCR assays.

The PCR reactions were carried out in 96-well plates (N8010560, Applied Biosystems™, Warrington, UK) and performed on a 7500 Fast Real-Time PCR System (Applied Biosystems™, Waltham, MA, USA) by adding 10 µL of PowerUp™, SYBR™ Green master mix (A25777, Applied Biosystems™, Waltham, MA, USA), 1 µL (10 µM) of each primer, and 2 µL of diluted cDNA. For amplification efficiency analysis, a standard curve of 7 points was included for each target gene consisting of 10-fold serial dilutions starting at 0.1 ng/µL of a conventionally prepared PCR product [39]. Data were analyzed by using the relative quantification 2^–ΔΔCt^ method as previously described [40]. Uninfected OvMØs were considered to produce basal levels of mRNA of all cytokines and iNOS investigated and thus considered as negative controls. Genes were considered differentially expressed when they presented a fold change of ≥2.

### 2.7. Statistical Analysis

For cIR, McR, cytokine, and iNOS mRNA expression levels, a statistical analysis was carried out that considered each isolate individually. However, in order to facilitate the interpretation and comparison between isolates with similar characteristics, *T. gondii* isolates were also divided and analyzed according to their origin (A: isolated from abortion or C: isolated from the myocardium of chronically infected adult sheep) and the genotype (PRU II, clonal type II, or type III), resulting in four groups as follows: PRU II-A: type II PRU—abortion origin (TgShSp1 and TgShSp3).PRU II-C: type II PRU—myocardium of chronic-infection origin (TgShSp11 and TgShSp16).Clonal II-A: clonal type II—abortion origin (TgShSp2).Type III-C: type III—myocardium of chronic-infection origin (TgShSp24).

The Kolmogorov-Smirnov test was used to assess the data normality. A Shapiro–Wilk test analysis was used to analyze the intra-experiment’s variability between replicates of the same isolate. On the other hand, the Friedman test was used to analyze the inter-experiment’s variability in the values within the same isolate between the different experiments. Finally, as the data were not normally distributed, the non-parametric Mann–Whitney test was performed to compare the cIR and McR levels observed and the cytokine and iNOS transcript levels between pairs of groups or isolates. The *p*-values < 0.05 were considered statistically significant. All statistical analyses were performed with GraphPad Prism 8.0.1 software (San Diego, CA, USA). 

## 3. Results

### 3.1. Cell Infection Rate and Multi-Infected Cell Rate

The internalization of the parasite into the OvMØs was expressed in the cIR and McR. There were no statistically significant differences (*p >* 0.05) between the results obtained in the different intra-experiments or inter-experiments assays (Appendix A). Significant differences were observed between all groups regarding the cIR (*p <* 0.05) (Figure 1 and Figure 2). Specifically, the highest values of the cIR were observed in the type III-C isolate (93.54% ± 6.46) followed by the clonal II-A isolate (79.96% ± 9.98), the PRU II-C group (71.60% ± 12.90), and finally the PRU II-A group (62.38% ± 15.76) (Figure 2). 

Concerning the McR results, the groups could be sorted from higher to lower in a similar order to that of the cIR: type III-C isolate (79.34 ± 13.50), PRU II-C group (46.59 ± 19.07), clonal II-A isolate (45.96 ± 20.44), and PRU II-A group (29.62 ± 16.63) (Figure 2). When comparing the data between groups, there were statistically significant differences in all cases except between the PRU II-C group and the clonal II-A isolate (Figure 2). 

Regarding the differences found when the isolates were considered individually and not grouped, the results are shown in Appendix A. Briefly, the highest cIR values were observed in the TgShSp24 and TgShSp2 isolates (93.51% ± 5.71 and 80.15 ± 8.91), and the lowest values were observed in the TgShSp1 (60.16% ± 12.71) (*p* < 0.001) and TgShSp3 isolates (64.92% ± 9.85) (*p* < 0.001), which was consistent with results of the group comparison. The McR values were similar to those observed in the cIR. The percentage of McR in the TgShSp24 isolate was the highest when compared to the other isolates (65.03% ± 4.59). There were no significant differences (*p* > 0.05) among the cell infection rates caused by the TgShSp3, TgShSp11, or TgShSp16 isolates; whereas TgShSp1 showed the lowest percentage of multi-infected cells (40.95% ± 12.69) (Appendix A, Appendix A).

### 3.2. Transcript Expression of Cytokines and iNOS

Regarding the analysis of the Th1 cytokines, a significant upregulation of IL-12 and TNF-α (*p <* 0.001) was observed regardless of the isolate compared to the negative controls. The IL-12 transcription was significantly higher in the type III-C (687.6-fold) and clonal II-A isolates (729.6-fold) compared with the PRU II-C (483.3-fold) and PRU II-A groups (270.5-fold) (*p <* 0.05) (Figure 3). Regarding the individual values, the TgShSp3 isolate (PRU II-A group) showed the lowest threshold increment in IL-12 transcription levels (108.1 ± 129.5) (Appendix A). Similarly, the transcription of TNF-α was significantly incremented in the clonal II-A isolate (38.53-fold; *p <* 0.05), the PRU II-C group (35.15-fold), and type III-C isolate (30.55-fold) when compared with the PRU II-A group (19.14 ± 5.1), but not between them. Specifically, the TgShSp3 isolate (PRU II-A group) showed the lowest TNF-α transcription levels (Appendix A). Interestingly, within the PRU II-C group there was a great difference (*p <* 0.05) between TgShSp11 and TgShSp16 isolates (21.78-fold and 48.53-fold, respectively) (Appendix A). 

The highest transcription of IL-6 was observed in the OvMØs infected with the clonal II-A isolate (74.19-fold), which showed statistical differences with all groups (*p <* 0.05) (Figure 3). When analyzing the individual isolates, the TgShSp11 isolate (PRU II-C group) (19.20-fold) and again the TgShSp3 isolate (PRU II-A group) (3.65-fold) showed the lowest increments (*p <* 0.05).

Concerning the analysis of the Th2 cytokines, the PRU II-A group presented the highest transcription level of IL-10 (3.05-fold); but significant differences (*p <* 0.05) were only found with the PRU II-C group. In contrast, the PRU II-A group showed the highest transcription level of IL-4 (3.79-fold) with statistical differences with the PRU II-C group and clonal II-A and type III-C isolates (Figure 3).

The IL-1α expression in the OvMØs showed a similar pattern to that of the Th2 cytokines; macrophages infected with the PRU II-A and PRU II-C groups were the only ones that showed an increase in transcription (5.15-fold and 3.74-fold, respectively) with regard to the clonal II-A and type III-C isolates (*p <* 0.001) (Figure 3). Similarly, when analyzing the transcription of IL-17, only the PRU II-C group (TgShSp11 and TgShSp16 isolates) showed a significant increase in transcription (31.96-fold, *p <* 0.05) when compared with the remaining groups (Figure 3 and Appendix A). Regarding iNOS expression, the higher fold-change values were observed in the PRU II-A group (3.41-fold) and the clonal II-A isolate (4.48-fold), whereas the expression of the remaining groups was similar to that of the uninfected control (*p* < 0.05) (Figure 3).

## 4. Discussion

The virulence of *T. gondii* isolates has been classically determined via the cumulative mortality rates in murine models, although recent studies on genetic and phenotypic variability of isolates have questioned this classification [16,41,42]. On the other hand, in vitro infection assays on immortalized or primary cell lines from rodents and humans have been relevant tools for the analysis of the phenotypic variability of *T. gondii* isolates [43], although only a few studies have used ruminant cells [22]. It is evident that the analysis of the interaction between immune cells and the parasite is of utmost importance in the study of the pathogenesis of toxoplasmosis. This interaction has been already addressed in several studies [44,45], some of which focused on ruminants [5]. In this sense, the analysis of MØs is widespread in the study of host–pathogen interactions for different intracellular organisms [11]. Furthermore, it is known that the genetic variability of *T. gondii* isolates relates to the type of response and polarization of macrophages [12]. However, there is still little information on the precise effect of *T. gondii* variability upon ovine macrophages—the most important immune target cell—despite the usefulness of the in vitro model of OvMØs being an excellent model to study the macrophage/pathogen interaction in relevant diseases of sheep [23]. 

In this work, the influence of a panel of six isolates, that have previously been characterized in a mouse model and in ovine trophoblast cell line [22], in the phenotypic variability on the response of OvMØs to the infection, was studied at 6 hpi, both in terms of the cIR and McR and in the transcript expression. At this time point, it was expected that the number of the tachyzoites was the same as when the cultures were infected because no multiplication of the parasite took place. This assumption was since all the isolates used in this study were expected to need 12 h to complete a chromosomal replication, as previously studied for types II and III, in contrast to type I strains that take 6–8 h [46]. In this study, we measured parasite internalization into macrophages, which could be either based on parasite active invasion or on macrophages actin-filament dependent phagocytosis [47]. Nevertheless, when a control for phagocytosis (heat inactivated tachyzoites) was performed, no differences in cIR values were observed between live and inactivated tachyzoites infection (data not shown) [33]. In addition, mRNA quantification and protein quantification were both used as appropriate techniques to study the changes in cellular function and pathways, as it has been stated that the presence of mRNA correlates with the levels of secreted proteins in many cases, particularly IFN-γ essential for *T. gondii* control [48]. However, in this study, the early immune response against *T. gondii* has been studied at 6 hpi, when the production of proteins in supernatants is too low due to requires around 21–24 hpi [49]. For that reason, protein quantification was not included. Bearing in mind the limitations of cytokine evaluation by mRNA expression because the final amount of cytokine secreted by the cell is unknown; we considered cytokine production could be proxied in this case by cytokine mRNA for comparisons between groups of isolates and for studying mechanistic pathways, as has been stated in other in vitro studies [50]; giving us an idea about the direction of travel of the macrophages towards the polarization.

The type III isolate (TgShSp24) showed the highest cIR and McR when compared to the type II isolates, and there were differences among them. These findings were similar to those of previous in vitro studies in which human monocytes (THP-1 cell line) were infected with type II and type III *T. gondii* isolates [51]. In fact, the type III isolate also showed a notable exponential growth in ovine trophoblasts (AH-1 cell line) [22]. The internalization into macrophages, specifically through phagocytosis, has been considered a strategy of low-virulence strains of *T. gondii* because the parasite can form a parasitophorous vacuole after phagocytosis and resist inside the cell [52]. This may lead to enhanced systemic dissemination in a trojan horse manner and favor the establishment of chronic infection [52]. Taking this into account, it remains to be studied whether this higher internalization rate of a type III isolate into OvMØs was an indication of higher virulence in an ovine in vivo model. On the other hand, it was previously reported that the reference ME49 isolate, a clonal type II strain, had a higher invasion rate in VERO cells than the reference VEG isolate, a clonal type III strain [53]. It should be noted that there were clear differences in the experimental design between these studies (e.g., VERO cells versus primary immune cells from sheep) in addition to the fact that it was previously demonstrated how *T. gondii* isolates can modify their phenotypic traits after a high number of passages in cell culture, as in the case of laboratory isolates such as ME49 [26]. For this reason, a panel of six *T. gondii* isolates that were recently obtained with a low number of cell-culture passages was used to carry out this phenotypic comparison.

The cell type used for the in vitro model could also influence the behavior of the isolates. The same PRU II-C group of isolates cultured in trophoblast (AH-1 cell line) showed higher cIR than the clonal II-A [22], while in the present study, the opposite occurred when cultured in OvMØs, which suggests that genetic variation on *T. gondii* isolates could determine different rates of invasion/proliferation depending on the cell population used. Recent studies have shown that the virulence of *T. gondii* isolates could depend, among other factors, on their capability to bind CD36 receptor in macrophages, as isolates of different virulence had different affinity for this receptor [54]. It was also interesting to find that there were differences in the cell infection rates between isolates of the same PRU-II genotype, depending on the origin of the isolates, that came from abortion or from chronically infected sheep. This could indicate a putative role of the origin of the isolate in the virulence of *T. gondii*. Interestingly, the study in which these isolates were characterized also found that at 8 hpi, a similar time point to the one used in the present investigation, isolates of the PRU II-C group showed a higher invasion rate than the PRU II-A group [22]. However, these results must be confirmed in further studies to evaluate whether the variation of phenotypic traits depending on the origin of the sample is a characteristic particular to the studied isolates or whether it can be extrapolated to other isolates.

The variability in the cIR and McR of the *T. gondii* isolates could also be reflected in the response of the macrophages to infection [55]. In our study, after infecting the OvMØs with *T. gondii* tachyzoites, a pro-inflammatory-cytokine M1 profile was directly related to a higher cIR and McR. This suggested that higher rates of internalization may help to polarize the macrophages into an M1 phenotype over M2, possibly with the aim of mounting a stronger response after a more aggressive infection. In this sense, a stronger Th1 response was found in those bovine MØs that were incapable of controlling a close pathogen/species *N. caninum* invasion [56]. In general terms, OvMØs infected with isolates from type III and clonal type II tend to differentiate into M1 macrophages with a higher transcription of IL-12 and TNF-α. It is well known that these cytokines have an essential role in controlling *T. gondii* proliferation because their inhibition in mice increased the severity of the disease [6]. IL-12 and TNF-α are key to the production of IFN-γ by NK and T cells, thereby giving rise to a robust a pro-inflammatory immune response in vivo, mediating host resistance against this parasite, and lowering the parasite burden [57,58,59]. When taking this into consideration, these results suggested that the initial contact of the OvMØs with tachyzoites from the clonal type II and type III isolates would trigger a pro-inflammatory response with the aim of controlling parasite multiplication. However, this does not necessarily involve protection against the disease because a strong pro-inflammatory response after *T. gondii* infection could also cause severe disease in the host [17]. To assess the implications of this response to the host, further in vivo infection experiments in a relevant host model are needed.

Similarly, OvMØs infected with clonal type II isolate showed an increased expression of IL-6, which corresponded with high cIR and McR values. This is a pleiotropic cytokine, that has pro- and anti-inflammatory functions and plays a key role in the protection against toxoplasmosis [60]. That one isolate with high cIR and McR also causes upregulation of IL-6 may also suggest, as the results discussed above, that higher internalization in OvMØs leads to the production of pro-inflammatory cytokines, or cytokines synergic with them [61]. The feature present in the clonal type II isolate that promotes IL-6 transcription remains unknown, and is probably not present in the rest of the isolates. 

On the other hand, OvMØs infected with PRUII-A and, to a lesser extent, with PRU II-C tachyzoites, tend to differentiate into a M2 profile, with an increase in IL-10, TGF-β and IL-4 mRNA expression. These cytokines are classically classified as anti-inflammatory with a role on inhibition of pro-inflammatory cytokines production [62,63,64,65]. Furthermore, they also block the ability of IFN-γ to activate macrophages, thus favoring the intracytoplasmic multiplication of the parasite [66]. However, it should be noticed that the threshold increase in these three cytokines is low, below 8 in IL-10 and IL-4. The question of whether macrophages would significantly contribute to the generation of these cytokines in *T. gondii* infection is raised because of their low increases. It is possible that other cells, including CD4+ T cells, are required to produce anti-inflammatory cytokines [67,68,69].

Whether macrophages infected with different isolates of *T. gondii* induce a pro- or anti-inflammatory microenvironment has been attributed to differences in the virulence factors of the parasite, such as dense granule or rhoptry proteins that could interact with host-dependent molecules such as NF-κB nuclear translocation factor or STATs signaling pathways. For example, previous studies have found that the infection of murine bone marrow-derived macrophages with *T. gondii* isolates lacking ROP16 promoted a M1 phenotype while those isolates with ROP16 protein induced a M2 phenotype [68,69]. Although ROP16 has not been studied in any of the isolates used in the current study, they do present differences in the *ROP5* and *ROP18* genes allelic profile, specifically between the type II and type III isolates [22]. Further genetic characterization of these isolates might help to elucidate which genes influence the response of the OvMØs to *T. gondii* infection. In the same way, the dense granule protein GRA15 from type II isolates could induce a M1 phenotype through activation of NF-κB [57,64,70]. The inoculation of human mononuclear cells with a type II isolate (ME49) induced the expression of TNF-α and IL-12 cytokines [71], and a similar response has been found in murine macrophages [72]. However, the relationship between the genetic background of the isolate and the response of infected cells is not always clear as infection of mice with type III isolates led to a M1 profile, and not the expected M2 [73]; furthermore, there were no differences in the cytokine profile of avian blood monocyte-derived macrophages after infection with type II isolate [74]. This variability regarding the relationship between parasite background and host response strongly suggests that a plethora of *T. gondii*-derived molecules, not only virulence factors such as ROP16 and GRA15, might be involved in the M1/M2 vias of macrophages. Even the same parasite molecule could cause opposite effects on the macrophages depending on its concentration, as citrate synthase I has been shown to either enhance or inhibit phagocytosis depending on the concentration [75].

The increase of IL-17 transcription in one group (PRU II-C) was an unexpected finding, as even thought this cytokine has been shown to participate in the control of murine toxoplasmosis [76,77], it is mainly produced by lymphocytes [78]. This increase in transcription may be associated with the higher mortality observed in mice infected with isolates from PRU II-C when studying their virulence [22], compared to the other isolates which did not result in an increase in IL-17 expression by macrophages. Its role in ovine toxoplasmosis is unknown, although the increase of this cytokine could be related to a protection against vertical transmission, as has been suggested in chronically infected sheep with the close-related parasite *N. caninum* [79]. It is also surprising that the observed increase of IL-1α expression, a pro-inflammatory cytokine-related protection against toxoplasmosis that drastically reduces parasite growth [80,81], found in OvMØs infected with PRU II isolates, was otherwise associated with a M2 profile. However, the role of IL-1α cytokines during *T. gondii* infection is not yet fully understood, as they have been shown to participate in the protection against toxoplasmosis in murine models but also to play no role or even to be counterproductive and favor the advance of the disease [82]. Regarding iNOS expression, those *T. gondii* isolates obtained from abortions (PRU II-A group and clonal II-A isolate) showed the highest values, which could suggest a greater capability to eliminate the parasite in vitro that need to be confirmed in in vivo studies. Altogether, these results indicate that the role of IL-17, IL-1α and iNOS on *T. gondii* infection is unclear when it comes to ovine toxoplasmosis and deserves further investigation.

In any case, the results of the present study, which showed different responses of the OvMØs to infection depending on the genetic background and origin of the *T. gondii* isolates, were limited to the in vitro conditions evaluated, so they cannot be easily extrapolated to the host immune response, in which a highly complex relationship between the parasite and a plethora of cell populations (not just macrophages) develops. Further studies in ovine experimental models may help to clarify the role that phenotypic variation in *T. gondii* isolates could play in the pathogenesis of ovine toxoplasmosis. In addition, these results in OvMØs were restricted to these Spanish *T. gondii* isolates and cannot be extrapolated to all type II and III isolates, which are expected to show intra-genotypic differences.

## 5. Conclusions

To conclude, the in vitro infection of OvMØs was shown to be a useful tool to phenotypically characterize *T. gondii* isolates. This study showed that there were inter- and intra-genotype variations in the parasite-OvMØs relationship for the recently obtained *T. gondii* isolates. Future investigations should address whether these variations in the parasite internalization and transcript expression could be extrapolated to the clinical course and pathogenesis of ovine toxoplasmosis by conducting in vivo animal studies.

## Figures and Tables

**Figure 1 animals-12-03453-f001:**
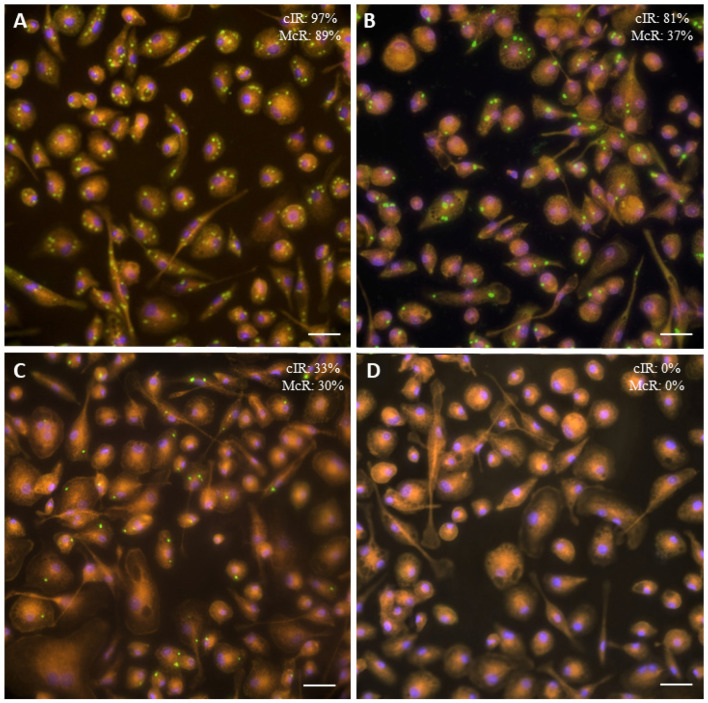
Immunofluorescence of OvMØs at different rates of infection with *Toxoplasma gondii* tachyzoites: (**A**) high rate (Type III isolate); (**B**) medium rate (PRU II-C group); (**C**) low rate (PRU II-A group); (**D**) uninfected OvMØs. *Toxoplasma gondii* tachyzoites (green—Alexa Fluor 488), cytoplasm (orange—TRITC), and nuclei (blue—DAPI) are shown. cIR: cell infection rate; McR: multi-infected cells. Scale bars = 50 μm.

**Figure 2 animals-12-03453-f002:**
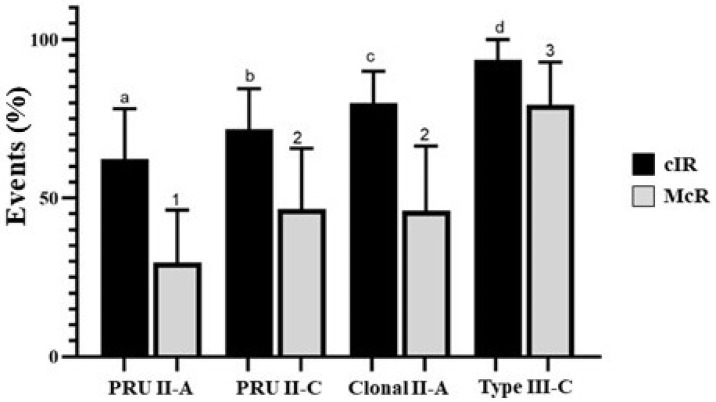
Cell infection rate (cIR) and multi-infected cell rate (McR) at 6 hpi in OvMØs for the four groups infected with *Toxoplasma gondii* isolates using a MOI of 3:1. The total number of cells, the number of infected cells, and the number of cells with multi-infection were determined via double immunofluorescence staining followed by counting using an inverted fluorescence microscope. Different superscript letters indicate statistically significant differences between pairs of the four groups of isolates regarding the cIR (Mann-Whitney test) (*p* ≤ 0.05). Pairwise comparisons (Mann-Whitney test) between the four groups of isolates for cIR and McR revealed statistically significant differences, as is indicated by the superscript letters and numbers, respectively.

**Figure 3 animals-12-03453-f003:**
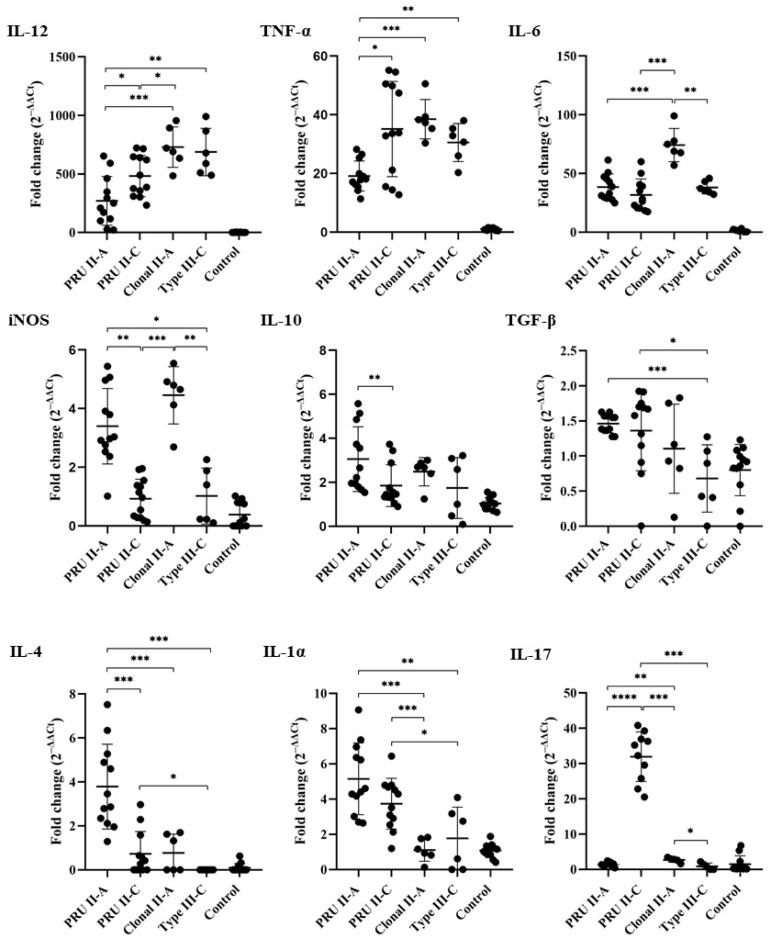
IL-12, TNF-α, IL-6, iNOS, IL-10, TGF-β, IL-4, IL-1α, and IL-17 transcript expression. Scatter plot graphs of relative mRNA expression levels (as a fold change) of cytokines and iNOS in OvMØs infected with different grouped *Toxoplasma gondii* isolates (MOI 3:1) at 6 hpi. PRU II-A: type II PRU–abortion origin (TgShSp1 and TgShSp3); PRU II-C: type II PRU–myocardium origin (TgShSp11 and TgShSp16); Clonal II-A: clonal type II–abortion origin (TgShSp2); Type III-C: type III–myocardium origin (TgShSp24). Data represent the mean and standard deviation. Asterisks over the bars indicate significant differences: * *p* < 0.05; ** *p* < 0.01; *** *p* < 0.001; **** *p* < 0.0001.

**Table 1 animals-12-03453-t001:** *Toxoplasma gondii* Spanish isolates selected for in vitro OvMØs characterization. All data were reported in previous studies [22,24].

Isolate	Type	Origin Clinical Sample	Genotype (ToxoDB)	Geographic Origin	In Vitro Model (AH1 Cell Line)		In Vivo Murine Model		
					Tachyzoite Yield 72 h (Zoites/ng of Total DNA)	Parasite Invasion Rate	Cumulative Mortality	Parasite Burden (30 dpi)	Clinical Sings
TgShSp1	Type II PRU variant	Ovine fetal brain	#3	Palencia, central Spain	44.5	Low	0%	Medium	Mild clinical signs (ruffled coat and ascites)
TgShSp2	Clonal Type II		#1	Navarra, northern Spain	58.4	Low	0%	Medium	
TgShSp3	Type II PRU variant		#3	Palencia, central Spain	112.5	Medium	0%	Low	
TgShSp11	Type II PRU variant	Adult myocardium of chronic infected sheep	#3	Cáceres, western Spain	128.8	Medium	8%	Medium	Rounded back Loss of body condition
TgShSp16	Type II PRU variant		#3	Badajoz, western Spain	97.9	Medium	20.8%	High	
TgShSp24	Type III		#2	Ciudad Real, central Spain	403.6	High (Exponential growth and larger vacuoles)	18.2%	High	

## Data Availability

Not applicable.

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
