# Peer review of "Effects of Ovine Monocyte-Derived Macrophage Infection by Recently Isolated Toxoplasma gondii Strains Showing Different Phenotypic Traits"

_animals, 2022, doi:10.3390/ani12243453_

Round 1

Reviewer 1 Report

I really appreciate your effort to perform these experiments. They were technically superior. However, I am disappointed that you have not included a mouse virulent strain in your study. Are the first 6 hours of infection so critical for infection? In the isolation of Toxoplasma gondii using the mouse, it often takes 6 weeks to check for infection. As you said, even the weak parasite can hitch a ride in the monocyte as a "Trojan horse" to various organs of the host. All your isolates are from the sheep, and so they are all proven, successful invaders. What I am aiming at is to have a simple test that can substitute for the mouse virulence test. I hope you can contribute more in this area.

Author Response

I really appreciate your effort to perform these experiments. They were technically superior. However, I am disappointed that you have not included a mouse virulent strain in your study. Are the first 6 hours of infection so critical for infection? In the isolation of Toxoplasma gondii using the mouse, it often takes 6 weeks to check for infection. As you said, even the weak parasite can hitch a ride in the monocyte as a "Trojan horse" to various organs of the host. All your isolates are from the sheep, and so they are all proven, successful invaders. What I am aiming at is to have a simple test that can substitute for the mouse virulence test. I hope you can contribute more in this area.

We appreciate the comments of the reviewer. We agree it is a good point of view include a mouse virulent strain. However, our work is based in a previous study (Fernández-Escobar et al., 2021) where six T. gondii isolates from sheep samples were obtained. The genotypic and phenotypic parameters of these isolates have been thoroughly characterized and these isolates are the most prevalent ones in sheep flocks of Europe (Fernández-Escobar et al., 2021). However, the prevalence of type I isolates (virulent strains in mouse) in Europe is scarce.

Taking this into account, the aim of our work was to analyse whether these isolates from the same host would behave differently in the same experimental model under the same experimental conditions, and specially how these isolates behave soon after infecting the cells. The time point of 6 hours was selected to in order to analyze the early immune response of macrophages once tachyzoites have been internalized in terms of cytokine transcript expression. However, when the production of cytokines in supernants want to be analyze, longer time periods are required (Mukhopadhyay et al., 2020)

In addition, a longer post infection time would have meant the possible multiplication of tachyzoites inside macrophages not allowing the initial internalization to be assessed correctly (Radke et al., 2001). We agree with the reviewer in that the analysis of longer periods of time would give relevant information into the pathogenesis of the disease but the aim of this work is to compare the early immune response in one of the most important immune target cells.

To address the issue that the reviewer mentions, we believe that an experimental in vivo infection of pregnant sheep would be more adequate. We do hope to perform such study in the near future and we believe that to compare the results from that study with the ones obtained in this in vitro model would add a lot of information into the characterization of T. gondii isolates and also into the pathogenesis of ovine toxoplasmosis.

We do concur with the reviewer in that a laboratory test able to characterize the virulence of T. gondii isolates, such a GWAS analysis or NGS approach would be a great achievement. However, we are afraid that there is still a lot to be done for the characterization of the phenotypic traits responsible for the isolate virulence and, what is more, how those traits influence the pathogenesis of the toxoplasmosis in the different hosts. While highly desirable, we reckon there are still many steps to be walked before such achievement could be reached. We humbly hope that our current study could contribute in some way towards that goal.

Fernández-Escobar M, Calero-Bernal R, Regidor-Cerrillo J, Vallejo R, Benavides J, Collantes-Fernández E, Ortega-Mora LM. In vivo and in vitro models show unexpected degrees of virulence among Toxoplasma gondii type II and III isolates from sheep. Vet Res. 2021 Jun 10;52(1):82.

Fernández-Escobar M, Schares G, Maksimov P, Joeres M, Ortega-Mora LM, Calero-Bernal R. Toxoplasma gondii Genotyping: A Closer Look Into Europe. Front Cell Infect Microbiol. 2022 Mar 23;12:842595. doi: 10.3389/fcimb.2022.842595.

Mukhopadhyay D, Saeij JPJ. Assays to Evaluate Toxoplasma-Macrophage Interactions. Methods Mol Biol. 2020;2071:347-370. doi: 10.1007/978-1-4939-9857-9_19.

Radke JR, Striepen B, Guerini MN, Jerome ME, Roos DS, White MW. Defining the cell cycle for the tachyzoite stage of Toxoplasma gondii. Mol Biochem Parasitol. 2001 Jul;115(2):165-75.

Reviewer 2 Report

reduce plagiarism

Accept after minor revision (corrections to minor methodological errors and text editing)

Author Response

After carefully examining the pdf attached by reviewer 2 which was extremely illustrative, we would like to emphasize that even a small percentage of the text overlap between our manuscript and earlier papers does not, in our opinion, constitute plagiarism. This occurs mostly in proper nouns, content that goes the same in all journal´s published articles, few words of a full phrase or sentences in the material and methods section because we have employed similar techniques from other studies of our research group. Please keep in mind that this article is a completely new study with distinct objectives and findings.

Reviewer 3 Report

The manuscript by Vallejo et al. 2022 entitled  Effects of recently isolated Toxoplasma gondii strains showing different phenotypic traits upon infection of ovine monocyte-derived macrophages; results from a simple but well strutured study addressing the parasite burden and immune response of in vitro infection of ovine monocyte-derived macrophages with Toxoplasma gondii isolates with different origins and genetic backgrounds.

The scientific data presented, seems to be solid and the experimental design was very simple but well conducted and the topic is in the scope of the journal section Small Ruminants. However, to be suitable for publication in Animals, it needs some modifications. 

Major concerns:

A very careful and MAJOR revision of the manuscript by a native English speaker should be performed to improve the level of standard English in order to make the manuscript clearer and easy to read, in particular in DISCUSSION section (but also in other sections including the title). Some examples:

Line 1-4 The title should be rephrased. My suggestion: "Effects of ovine monocyte-derived macrophages infection by recently isolated Toxoplasma gondii strains showing different phenotypic traits. 

Line 21 “ higher number of tachyzoites inside macrophages cytoplasm ” should be replaced by "higher internalization/infection rate".

Lines 23-25 The last sentence should be rephrased as the words "differences/different" appear multiple times and a conclusion sentence that should be more robust and sounding. 

Line 32 “iNOS” should be full written, as is the 1st time it appears in the text. 

In my opinion the following sentences should be rephrased/replaced in order to improve the english and/or make them less complex, more clear and easy to read and understand:

Lines 34-37; 86-89; 223-224; 288; 316-317; 345-348; 355-356; 387-388; 416-420; 432-436; 449; 465-467; 

Table 1 should be properly formatted (e.g: "Cumulative")

In Figure 2 the legend of the y-axis is missing. 

Line 310 The authors should mention which Figure is the sentence related. 

Line 400 The word "characteristic" should be deleted.

Line 412 The word "warranted" should be deleted.

Line 423 The word "cytokines" should be deleted.

Line 438 The word "isolates" should be corrected.

Line 449 "not the M2 expected" should be corrected to "not the expected M2"

Other major concerns:

The authors should show the graph of previous in vitro infection of OvMos with T. gondii (1, 2, 4, 6 and 24 hpi) as suplementary. 

It is not correct to use non-parametric Mann-Whitney test to multiple groups. The correct test is the nonparametric Kruskal–Wallis test , followed by Dunn’s multiple comparison test. 

In the DISCUSSION section, the authors mentioned that it was unexpected to find increased IL-17 mRNA only in PRU II-C group. In my opinion, it might not be as that unexpected since this cytokine is mainly produced by the Th17 cell subset. The authors might be discuss this issue. 

The authors should address the reason(s) for not quantify those cytokines and NOS in the culture supernatants and the limitations/advantages of mRNA quantification vs protein quantification. 

In the CONCLUSIONS section, the authors should point some of the future research that should be made taking into account the current results. 

Author Response

The manuscript by Vallejo et al. 2022 entitled “Effects of recently isolated Toxoplasma gondii strains showing different phenotypic traits upon infection of ovine monocyte-derived macrophages”; results from a simple but well strutured study addressing the parasite burden and immune response of in vitro infection of ovine monocyte-derived macrophages with Toxoplasma gondii isolates with different origins and genetic backgrounds.

The scientific data presented, seems to be solid and the experimental design was very simple but well conducted and the topic is in the scope of the journal section Small Ruminants. However, to be suitable for publication in Animals, it needs some modifications. 

We would like to thank the reviewer for the comments on the manuscript.

Major concerns:

A very careful and MAJOR revision of the manuscript by a native English speaker should be performed to improve the level of standard English in order to make the manuscript clearer and easy to read, in particular in DISCUSSION section (but also in other sections including the title). Some examples:

Following the indications of the reviewer, the manuscript has been reviewed by a native speaker (see certificate attached)

Line 1-4 The title should be rephrased. My suggestion: "Effects of ovine monocyte-derived macrophages infection by recently isolated Toxoplasma gondii strains showing different phenotypic traits. 

Done

Line 21 “higher number of tachyzoites inside macrophages cytoplasm” should be replaced by "higher internalization/infection rate".

Done

Lines 23-25 The last sentence should be rephrased as the words "differences/different" appear multiple times and a conclusion sentence that should be more robust and sounding. 

Done

Line 32 “iNOS” should be full written, as is the 1st time it appears in the text. 

Done (Line 32)

In my opinion the following sentences should be rephrased/replaced in order to improve the english and/or make them less complex, more clear and easy to read and understand:

Lines 34-37; 86-89; 223-224; 288; 316-317; 345-348; 355-356; 387-388; 416-420; 432-436; 449; 465-467; 

This has been addressed by the language reviewer

Table 1 should be properly formatted (e.g: "Cumulative")

Done

In Figure 2 the legend of the y-axis is missing. 

Done

Line 310 The authors should mention which Figure is the sentence related. 

Done, it is mention in lines (310-311)

Line 400 The word "characteristic" should be deleted.

Done

Line 412 The word "warranted" should be deleted.

Done

Line 423 The word "cytokines" should be deleted.

Done

Line 438 The word "isolates" should be corrected.

Done

Line 449 "not the M2 expected" should be corrected to "not the expected M2"

Done

Other major concerns:

The authors should show the graph of previous in vitro infection of OvMos with T. gondii (1, 2, 4, 6 and 24 hpi) as suplementary.

New graphs have been included as Supplementary figures

It is not correct to use non-parametric Mann-Whitney test to multiple groups. The correct test is the nonparametric Kruskal–Wallis test, followed by Dunn’s multiple comparison test.

Certainly, it is much more accurate to use Dunn’s multiple comparison test for comparison between groups. Dunn´s multiple comparison test was also performed and gave us similar results. However, we decided to leave Mann-Whitney test as we were comparing each group with another (pairwise comparisons) and it gave us a higher number of statistically significant results.

In the DISCUSSION section, the authors mentioned that it was unexpected to find increased IL-17 mRNA only in PRU II-C group. In my opinion, it might not be as that unexpected since this cytokine is mainly produced by the Th17 cell subset. The authors might be discussed this issue.

We are really grateful for this observation as the reviewer is absolutely right and it was not understandable in the way it was written on the manuscript. We have made the changes we considered necessary to improve this section of the discussion (Lines 472-477).

“The increase of IL-17 transcription on one group (PRU II-C), is an unexpected finding, as even thought this cytokine has been shown to participate in the control of murine toxoplasmosis [80,81], it is mainly produced by lymhocytes [82]. These increase of transcription may be associated to a higher mortality observed in mice in the isolates from PRU II-C when studying their virulence [25], compared with the remaining isolates which did not cause increase in IL-17 expression by macrophages”

What was meant to be said is that it is unexpected that this cytokine is express only by one group and not by the others (i.e It is not unexpected that the other groups did not express this cytokine). Even the main producer of IL-17 A are not macrophages, there are studies in where the production of IL-17 has been associated with macrophages and neutrophils:

Vykhovanets EV, Maclennan GT, Vykhovanets OV, Gupta S. IL-17 Expression by macrophages is associated with proliferative inflammatory atrophy lesions in prostate cancer patients. Int J Clin Exp Pathol. 2011 Aug 15;4(6):552-65.

Zhu X, Mulcahy LA, Mohammed RA, Lee AH, Franks HA, Kilpatrick L, Yilmazer A, Paish EC, Ellis IO, Patel PM, Jackson AM. IL-17 expression by breast-cancer-associated macrophages: IL-17 promotes invasiveness of breast cancer cell lines. Breast Cancer Res. 2008;10(6).

Kim JS, Jordan MS. Diversity of IL-17-producing T lymphocytes. Cell Mol Life Sci. 2013 Jul;70(13):2271-90.

The authors should address the reason(s) for not quantify those cytokines and iNOS in the culture supernatants and the limitations/advantages of mRNA quantification vs protein quantification

We agree with the reviewer that it is necessary to discuss this point. The main reason for not analyzing culture supernatants was that the time post infection when cells were collected, i.e. 6h, was too soon to properly evaluate the production of cytokines, so the best approach in order to study the polarization of the cells was to analyse the transcription of RNA. This information has been added to Discussion section (Lines 367-369).

“In addition, mRNA quantification and protein quantification are both used as appropriate techniques to study the changes in cellular function and pathways, as it has been stated that the presence of mRNA correlates with the levels of secreted proteins in many cases, particularly IFN-γ essential for T. gondii control [52]. However, in this study, the early immune response against T. gondii has been studied at 6 hpi, when the production of proteins in supernatants is too low due to requires around 21-24 hpi [53]. For that reason, protein quantification was not included. Bearing in mind the limitations of cytokines evaluation by mRNA expression because the final amount of cytokine secreted by the cell is unknown; we considered cytokine production could be in this case proxied by cytokine mRNA for comparisons between groups of isolates and studying mechanistic pathways, as it has been stated in other in vitro studies [54]; giving us an idea towards which polarization may macrophages go”

In the CONCLUSIONS section, the authors should point some of the future research that should be made taking into account the current results. 

Done as requested by the reviewer (Line 505-507)

“Future investigations should address whether these variations on the parasite internalization and transcript expression could be extrapolated to the clinical course and pathogenesis of ovine toxoplasmosis by conducting in vivo animal studies”

Round 2

Reviewer 3 Report

Thank you so much for addressing all the issues I raised. 

Best regards